# Leveraging Snow Probe Data, LiDAR, and Machine Learning for Snow Depth Estimation in Complex Terrain Environments

Dane Liljestrand[1], Ryan Johnson[1], Bethany Neilson[2], Patrick Strong[2], and Elizabeth Cotter[1]

[1]Department of Civil and Environmental Engineering, University of Utah, Salt Lake City, 84112, UT, USA
[2]Department of Civil and Environmental Engineering, Utah State University, Logan, 84322, UT, USA

**Correspondence:** Dane Liljestrand (dane.liljestrand@utah.edu)

**Abstract.** The majority of the water supply for many Western U.S. states is derived from seasonal snowmelt in mountainous regions. This study aims to generate basin-scale snow depth estimates using a multi-step, Gaussian-based machine learning model that combines snow probe depth measurements with static LiDAR terrain features from a single snow-free date, enabling rapid, high-resolution estimation at low institutional cost. We focus on reducing personnel danger by modifying the algorithm to minimize the exposure of field sample collectors to avalanche-prone terrain. Using snow observations taken solely within a subbasin ($\sim$9-$km^2$) of a larger basin ($\sim$70-$km^2$), a basin-scale snow depth estimate is modeled for a given date throughout the snow season. Results show that a small number of observations (i.e., 10) within a subbasin can realize snow depth across the greater basin with high accuracy, with a root mean squared error (RMSE) of 0.37 m, and Kling-Gupta efficiency (KGE) of 0.59 when compared to LiDAR snow depth distribution. We test the universality of the algorithm by modeling multiple subbasins of differing spatial characteristics and find similar results. The algorithm shows consistent performance across subbasins with varying spatial characteristics and maintains accuracy even when high-risk avalanche areas are excluded from the training data. This method exhibits a potential for citizen-scientist data to safely provide gridded modeled snow depth across different spatial ranges in snow-covered basins.

## 1   Introduction

Seasonal snow-derived water is a critical component of the water supply in mountainous basins and connected downstream regions (Painter et al., 2016; Bales et al., 2006). More than one-sixth of the world's population is in a region where snowmelt accounts for at least 50% of the annual runoff (Barnett et al., 2005). In the Western United States, where the economic value of the yearly snowpack has been estimated to be on the order of a trillion US dollars (Sturm et al., 2017), many states' water supply is nearly entirely dependent on mountain snowpack. Climate variability is pressuring this water supply and over the last century, much of the West has observed decreasing available water from snow, and more rapidly over the past 20 years (Mote et al., 2005, 2018).

Increasing populations and changing climate dynamics outline the crucial endeavor of accurately measuring available seasonal snow for water resource management. Acknowledging this need, the U.S. Natural Resource Conservation Service has installed and operates nearly 900 snow telemetry (SNOTEL) in situ monitoring sites throughout the Western U.S. (NRCS,

2022). These stations maintain the largest near-instantaneous monitoring network of snow depth and other environmental variables, forming the foundation for many water resource management forecasts throughout the country. Despite the broad network, the full spatial representation of mountainous regions remains a problem. Within the contiguous U.S., there is on average 1 SNOTEL site per approximately 4,000 km$^2$ of potential snow-covered area (Rutgers University Global Snow Lab, 2023). The low density of sites highlights the need for additional observations or techniques to produce an accurate and continuous
snowpack estimate.

Accurately representing basin-scale (30-200 km$^2$) snowpack is challenging due to scale and geographic heterogeneity. A single basin can exhibit broad slope, elevation, and land cover differences, all influencing snow distribution. While SNOTEL serves as the largest in situ data set in the U.S., studies have found that station measurements often did not align with mean values from surrounding areas, and vary in their broader accuracy during accumulation and melt seasons (Molotch and Bales,
2005; Lundquist et al., 2005; Meromy et al., 2013; Herbert et al., 2024). Heterogeneity in snowpack arises from interactions between the landscape features, wind, and forest canopy, among other factors, with varying effects across scales (Clark et al. (2011)). While small-scale processes like wind and radiative fluxes dominate at the hillslope scale (1-100 m), elevation becomes crucial at larger scales. Often, snow depth may be well-represented by a Gaussian distribution. However, the distribution is commonly skewed during accumulation and melt periods, in wind-affected terrain, or when no-snow areas are present, and a
static probability density should not always be assumed (He et al., 2019; Ohara et al., 2024). The multitude of factors affecting snow depth at varying scales make it challenging to universally identify the most relevant features for accurate snow depth estimation and complicate representative sampling.

When attempting to quantify a snowpack through measurement locations—whether for permanent instrumentation or one-time sampling—it is important to optimize the placement to represent various physical features effectively. Pioneering work
on optimizing snow measurement networks was performed by Galeati et al. (1986), who applied a multivariate statistical methodology with the aim of selecting a reduced number of monitoring stations within the Italian Alps monitoring network. The study applied a clustering technique on station observations and performed principal component analysis to determine insignificant or redundant stations. Their results showed that 30% of the network stations could be removed and suitably replaced with observations from neighboring stations.

While early studies focused on optimizing existing networks, recent research has enhanced network efficiency by optimizing measurement locations before installation. These studies have shown that fewer, strategically placed sensors can reduce modeled snow error in a relatively small, monitored catchment (Collados-Lara et al., 2020; Kerkez et al., 2012; López-Moreno et al., 2011; Oroza et al., 2016; Saghafian et al., 2016; Welch et al., 2013). Despite these advancements, challenges remain in the resource-intensive task of physically locating and installing measurement stations, as well as the fact that static locations
may not be ideal throughout different phases of the snow cycle. Additionally, the representativeness of point measurements over larger areas remains uncertain, and the extent to which these measurements accurately reflect snowpack conditions beyond their immediate vicinity is unknown.

Remote sensing products such as Snow Data Assimilation System (SNODAS) offer an alternative to ground-based measurement networks by providing snowpack estimates over large areas using a combination of ground-based, airborne, and satellite

observations (Barrett, 2003). However, SNODAS and similar products have their limitations, including relatively coarse spatial resolution (1 km), which can miss fine-scale variability in snow depth, and inaccuracies in complex terrains or densely forested areas where ground-based observations are sparse (Clow et al., 2012).

Aerial light detection and ranging (LiDAR) has improved the feasibility of high-resolution capture of snow depth data without relying on ground-based measurement stations and the spatial constraints of traditional remote sensing products. By subtracting a baseline snow-free DEM from a LiDAR-derived snow-on DEM, the resulting difference between the two precisely measures the snow depth across the surveyed area (Deems et al., 2013). While LiDAR is particularly effective in regions with deeper, uniform snowpack, it has limitations, and performance diminishes under canopy interactions and in shallower snow regions (Harder et al., 2016). The financial costs of LiDAR instrumentation and acquisition, especially for repeat or large-area surveys, also pose a significant constraint. Additionally, flying during adverse weather conditions makes monitoring snowpack changes throughout the season difficult. Contrastly, LiDAR surveys of snow-free terrain, which are less temporally constrained, can be conducted during the non-snow season to capture static surface features. To enhance the accuracy of regional snowpack estimates, physiographic surface data from snow-free LiDAR scans can be combined with on-ground point measurements of snow. This technique integrates the detailed localized data from ground observations with the broad coverage offered by LiDAR. When optimally located, the point measurements may help refine snow depth estimates and improve the overall understanding of snowpack variability (Oroza et al., 2016).

An increasing number of recreationists and citizen scientists in remote snow-covered environments provides opportunities for numerous low-institutional-cost point measurements across different spatial and temporal ranges. On-ground snow depth data reported by such users via a mobile app platform (details at communitysnowobs.org) provides a novel data source for scientists and water managers to supplement higher-cost collection methods (Crumley et al., 2021). However, access to remote sampling locations often can require a researcher or recreationist to travel in, under, or above avalanche terrain, exposing them to a potentially fatal outcome (CAIC, 2022). Nearly all avalanche fatalities occur in remote, uncontrolled terrain, with the majority occurring from individuals caught in a self-triggered avalanche or by another member of their group (Techel et al., 2016; Schweizer and Lütschg, 2001). Thus, it is imperative to develop sampling methods in remote regions that address and protect the safety of the individuals collecting data.

For snow sampling and modeling, snow water equivalent (SWE) is the most critical variable for predicting runoff and downstream hydrological processes (Clow et al., 2012). Because SWE is a function of both snow depth and density, improved estimates of either parameter can enhance SWE accuracy. Of these, snow depth is more readily and repeatedly measured during single-day field surveys and is also the primary variable captured by LiDAR. In contrast, snow density is more difficult to measure and requires more time and effort. Where available, snow surveys provide in situ point estimates of snow density, while parameterized models or generalized classifications provide more spatially extensive estimates. By combining high-resolution snow depth data with modeled or interpolated density values, researchers may generate more accurate spatial SWE estimates (Jonas et al., 2009; Sturm et al., 2010; Sturm and Liston, 2021).

In this study, we utilize a multi-step, Gaussian-based, machine learning model to investigate the feasibility of generating rapid, high-resolution, basin-scale snow depth estimates by combining snow probe depth observations with (static) LiDAR ter-

rain features with a built-in reduction in personnel danger from avalanche exposure. Within this work, we address three main objectives: i) validate the model's universality by sampling separate subbasins with differing spatial characteristics using a limited number of in-situ measurements to estimate snow depth and evaluate the performance in varied, complex terrain; ii) investigate the accuracy of basin-scale estimation beyond a smaller sampling domain with sparse sampling locations both within and outside of the modeled basin; and iii) determine if estimation accuracy is affected by the exclusion of high-avalanche-risk terrain when selecting measurement locations.

## 2 Materials and Methods

### 2.1 Study Area Description

We focus our study on the Franklin Basin region at the Utah-Idaho border, which encompasses the headwaters of the Logan River and its upper tributaries. The Logan River supplies the major population of the Cache Valley, with an average annual flow of approximately 6.5 cms at the mouth of the canyon, with a snowmelt-dominated hydrograph that delivers peak flow in the spring (Neilson et al., 2020). The limits of the study span an area within the Bear River Range of the greater Western Rocky Mountains (Figure 1). The study basin's elevation ranges from 2115 to 2940 m, with a mean of 2530 m, and it is predominantly easterly-facing. It is vegetated primarily with forest, range land, and alpine environments at upper elevations. The geology of the basin is characterized primarily by limestone and dolomite (Dover, 1995). Little development exists within this region, with the primary infrastructure consisting of a local ski resort, forest access roads, and seasonal homes. We select three subbasins to study and compare with the overall extent of Franklin Basin: Hell's Kitchen Canyon, which neighbors the larger basin to the south, and Boss Canyon and Peterson Hollow, both located centrally within Franklin Basin. All three sub-watersheds drain to the Logan River.

Hell's Kitchen Canyon is the southernmost subbasin in the study area and a popular recreation area throughout winter and summer. It is the lowest elevation of the three subbasin study areas (Table 1). Aspects are mainly northerly, southerly, and easterly facing with an overall east-facing aspect. A small area to the east of the canyon watershed is included in the study area to include more westerly-facing aspects during sampling and model training (Figure 1). The Boss Canyon subbasin is located north of Hell's Kitchen Canyon with mainly northerly, southerly, and easterly facing aspects and is an east-facing catchment. Peterson Hollow is a lower elevation catchment within the region and a predominately southernly facing drainage. Vegetation variation is similar across all subbasins, with each area predominately consisting of forested evergreen, aspen, shrubland, and open-range areas. The furthest western areas of Boss Canyon and Franklin Basin contain high-elevation, steep, sparsely vegetated slopes on the eastern aspect of the Wasatch Range Ridge.

### 2.2 Data Collection and Pre-Processing

We collected snow-free LiDAR data of the Franklin Basin in the fall of 2020. The collection was performed with an Optech Galaxy T2000 and Prime onboard a small aircraft at an altitude of 1300 m. An average flight density of 10.1 points per square

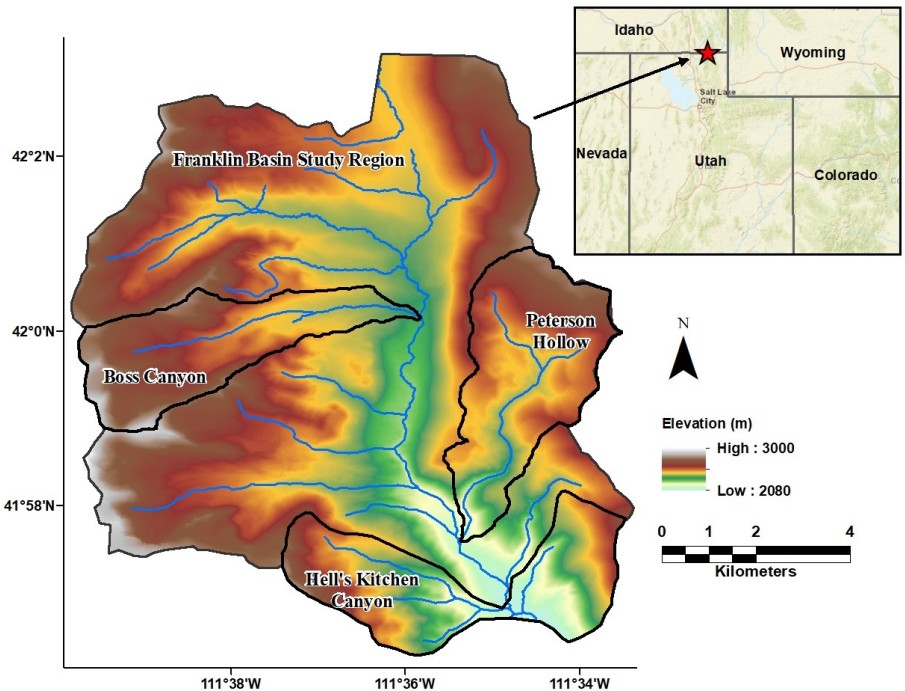

**Figure 1.** Map of the Franklin Basin study area extents and elevation and the Boss Canyon, Peterson Hollow, and Hell's Kitchen Canyon subject subbasins in Northern Utah and Southern Idaho, US (Inset map: ESRI).

**Table 1.** Topographic statistics of the four study study area.

| Domain | Area $(km^2)$ | Min El. (m) | Max El. | Mean El. | % Avalanche Terrain |
|---|---|---|---|---|---|
| Franklin Basin | 72 | 2115 | 2940 | 2530 | 5.2 |
| Hell's Kitchen Canyon | 8.9 | 2077 | 2830 | 2400 | 9.7 |
| Boss Canyon | 7.7 | 2320 | 2935 | 2613 | 11.6 |
| Peterson Hollow | 9.3 | 2164 | 2806 | 2520 | 2.3 |

meter was captured with a stated sensor absolute vertical accuracy of <0.03-0.25 m RMSE from 150-6000 m above ground level. The data is referenced to the NAVD88 and NAD83 vertical and horizontal datums and reprojected from State Plane Utah North to the WGS 84 / UTM Zone 12 coordinate system. We derived digital elevation model (DEM) and digital surface model (DSM) rasters from the obtained LiDAR and upscaled the resolution from its native resolution of 1.5 m to a 50 m grid cell

size with bi-linear interpolation to reduce computational demand (Table 2). The rasters for Hell's Kitchen Canyon were not upscaled, as this subbasin served as the field measurement boundary, requiring higher resolution to accurately guide samplers to precise sampling locations. The distribution of snow depth in the basin follows an approximately Gaussian distribution (Figure 3) with a mean and standard deviation very close to a theoretical Gaussian distribution, and a low Kolmogorov-Smirnov statistic (Eq. 10) of 0.06.

**Table 2.** Metadata summary of snow depth and terrain datasets. In Franklin Basin, Boss Canyon, and Peterson Hollow, raster data were upscaled to improve computational efficiency. Hell's Kitchen Canyon raster data were maintained at 1.5m for accurate locating of physical sampling locations.

| Dataset | Source | Resolution | Coverage |
|---|---|---|---|
| LiDAR Snow Depth | Airborne LiDAR | 1.5 m (raw), 50 m (upscaled) | Franklin Basin, Boss Canyon, Peterson Hollow |
| DEM & DSM | Airborne LiDAR | 50 m (upscaled) | Franklin Basin, Boss Canyon, Peterson Hollow |
| In-Situ Snow Depth | Manual Probes | Point-based (10 sites) | Hell's Kitchen Canyon |
| DEM & DSM | Airborne LiDAR | 1.5 m | Hell's Kitchen Canyon |

We then extracted the physiographic parameters of slope, aspect, and canopy height (Figure 2). We combine the slope and aspect grid values to calculate northness and eastness metrics with the equation,

$$northness = sin(slope) * cos(aspect)$$
$$eastness = sin(slope) * sin(aspect)$$

(1)

The values of northness and eastness both range from -1 to 1. In the northern hemisphere, a northness value of -1 corresponds to a steep, southerly-facing slope, while a value of 1 indicates a steep, northerly slope. Similarly, an eastness value of -1 corresponds to a steep, west-facing slope, and a value of 1 corresponds to a steep, east-facing slope (Collados-Lara et al., 2017; Fassnacht et al., 2003). Both northness and eastness can be interpreted as proxies for exposure to solar radiation (Amatulli et al., 2018).

We derive a wind shelter metric for the study domains according to the method of Winstral et al. (2002), quantifying the degree of shelter/exposure provided by upwind terrain. For each cell in the raster, a search distance of 100 m of the DEM in the northerly direction - the prevailing wind direction in the region - was applied (Western Regional Climate Center, 2022). As a final pre-processing step, we normalize the physiographic features from 0 to 1 with a min-max scaler to improve the stability of the model during learning.

To identify avalanche-prone terrain in the region, we inspect the average slope angle of each raster cell. Cells with a slope angle of 30° or greater are defined as having potential avalanche risk. For slopes below 30°, gravitational forces lack the strength to initiate a slide avalanche, so we consider these areas non-avalanche-prone (Maggioni and Gruber, 2003). The potential for avalanches to trigger on adjacent slopes or to progress to slopes below 30° exists, however was not included in our terrain assessment. From the classification, we create two data frames for the feature space: one includes all-terrain cells, and the other excludes cells marked as avalanche-prone, allowing us to compare them during model evaluation.

On Mar 28, 2021, near the end of the accumulation season, the flight crew collected snow-on LiDAR data for Franklin Basin with the same instrumentation as the snow-free LiDAR. With the collected LiDAR, we developed a snow depth TIFF for the basin by raster subtracting the bare earth DEM from the snow-on DEM. The result is a normally distributed snowpack across the basin, with a mean depth of 1.28 m and a standard deviation of 0.44 m. Due to flight pattern constraints, snow-on LiDAR was not captured for the Hell's Kitchen area. The Franklin Basin, Boss Canyon, and Peterson Hollow raster resolution were

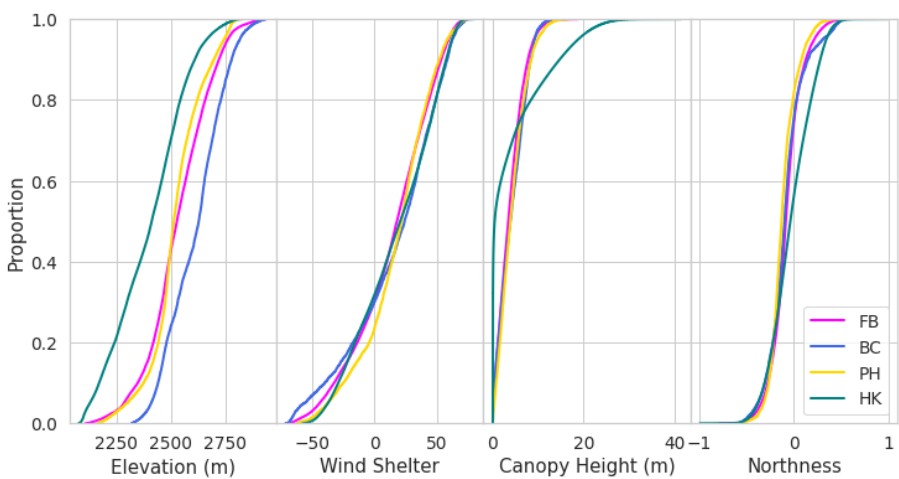

**Figure 2.** Cumulative distributions of the physiographic features for each region domain. FB indicates Franklin Basin, BC: Boss Canyon, PH: Peterson Hollow, and HK: Hell's Kitchen

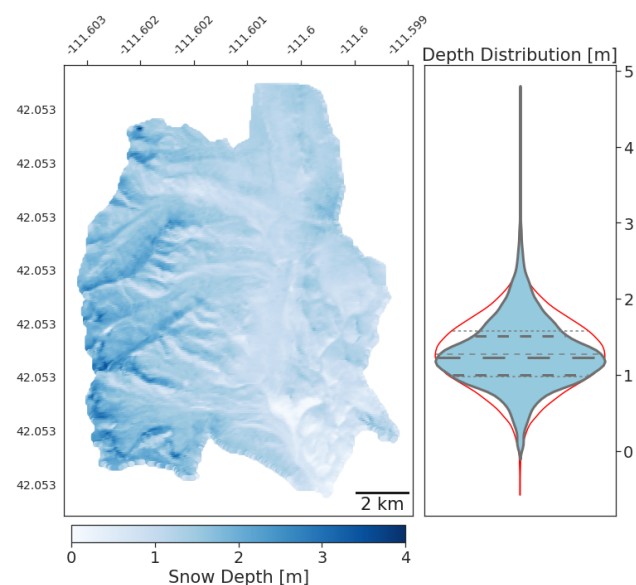

**Figure 3.** Map and corresponding density distribution of the LiDAR-measured snow depth in Franklin Basin. The violin distribution of the snow depth is overlayed above a Gaussian distribution of 20000 randomly generated samples to express the near-normality of the snow depth distribution. Quartiles of the snow depth are shown as bold hashed lines, and the light grey hashed lines represent the Gaussian distribution quartiles.

upscaled to 50 m to reduce computation time. In comparison, the Hell's Kitchen Canyon bare earth DEM was maintained at a
1.5 m resolution to provide detailed locations for the physical sampling.

## 2.3    Optimal Measurement Location Identification

To optimally identify field sampling locations to produce snow depth estimates we apply a multi-step, Gaussian-based machine learning algorithm (Figure 4). The framework builds on previous work, which found the algorithm successful in capturing snow depth variability within a basin with a small number of optimized sensors (Oroza et al., 2016). Our procedure involves using

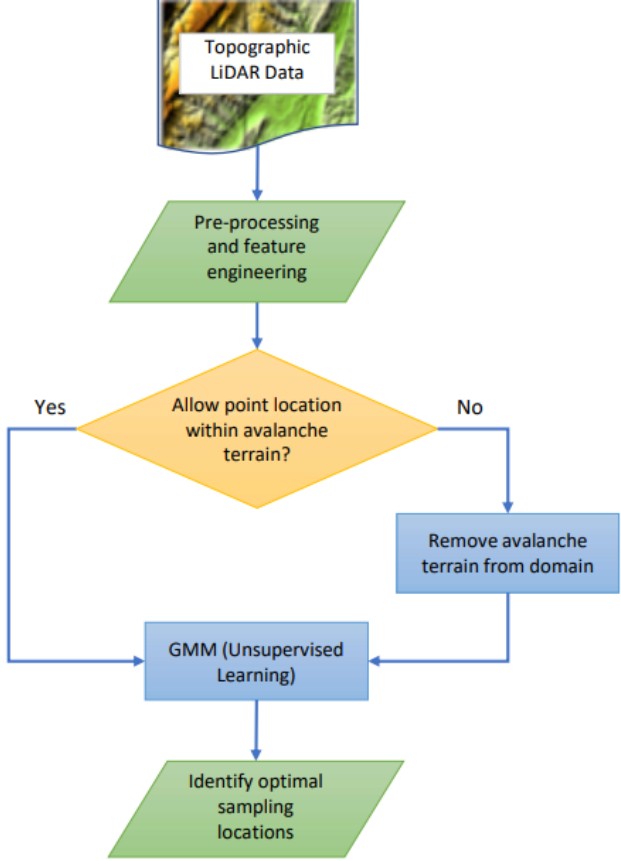

**Figure 4.** Workflow of the optimal sample locating model. The model relies on the GMM to find the most physiographic representative points in a domain while considering avalanche-prone terrain.

an unsupervised Gaussian Mixture Model (GMM) to identify sampling locations within Franklin Basin and its subbasins that best represent the region's physiographic composition based on snow-free topographic features (DEM elevation, northness, eastness, canopy height, wind shelter). Unsupervised learning, like the GMM, requires no dependent variable input to identify these representative data points.

The GMM is a probabilistic model that characterizes the feature space as a composition of several Gaussian distributed clusters, $K$, with mixing coefficients $\pi$, such that,

$$\sum_{k=1}^{K} (\pi_k) = 1. \tag{2}$$

Each cluster is defined by a Gaussian density function (Eq. 3), of D dimensions, expected value, $\mu$ and covariance, $\sigma$, and where $\mathbf{x}$ represents the snow depth. The multivariate distribution is expressed as,

$$\mathcal{N}(\mathbf{x} \mid \mu, \sigma) = \frac{1}{(2\pi)^{D/2} |\sigma|^{1/2}} \exp\left(-\frac{1}{2}(\mathbf{x} - \mu)^T \sigma^{-1} (\mathbf{x} - \mu)\right). \tag{3}$$

The optimal $\mu$ and $\sigma$ parameters for a given distribution can be found by taking the log of Eq. 3, differentiating, and equating it to zero. For multiple Gaussian distributions, optimal parameters are determined by maximizing the log-likelihood of all components over the entire feature space for a range of points, $N$. The log-likelihood of the GMM is defined as,

$$\ln p(\mathbf{X}) = \sum_{n=1}^{N} \ln \sum_{k=1}^{K} \pi_k \mathcal{N}(\mathbf{x}_n \mid \mu_k, \sigma_k). \tag{4}$$

We used the open-source Scikit-Learn Python library to execute the Gaussian mixture-based multi-step optimization model (Pedregosa et al., 2011). To locate the optimal parameters for the dataset, the GMM employs the Expectation-Maximization (EM) algorithm (Dempster et al., 1977), given a specified number of clusters. The EM algorithm iteratively adjusts parameters for the mixture of components until it arrives at a maximization of the log-likelihood function, thus defining the most representative feature points within the dataset. To avoid convergence on local maxima, we run a grid search of randomized algorithm initializations with either 10 or 50 restarts to select the result that maximizes the log-marginal likelihood. We calculate with a spherical covariance kernel from a random seed of initial cluster origins and sub-sample 80% of the domain for computational efficiency.

We defined the total number of clusters (sampling sites) as 10 to focus the study on sampling ability. 10 sites allow field measurement collectors enough time to traverse to all locations within a single day of sampling. To ensure the limited number of sites does not adversely impact model performance, we test the sensitivity of the model to the number of training sites by executing multiple instances. For each of the Franklin Basin Region, Boss Canyon, and Peterson Hollow areas, we run the model initially with 5 clusters and increase the number of clusters with each iteration to 100. For each cluster center defined by the GMM, a ball-tree nearest neighbor search method is applied to locate the cell location most closely represented by the features within the feature space. The output of the 10 GMM-identified optimal locations within the topographic feature domains of Franklin Basin, Boss Canyon, and Peterson Hollow is shown in Figure 5. Within the Hell's Kitchen Canyon subbasin, we performed a single model execution of 10 clusters to locate 10 optimal sampling sites for field measurement.

We run the GMM twice for each feature space: once including cells defined as avalanche-prone and once excluding them. In each scenario, we use the same GMM with nearest neighbor search method with an identical number of training sites. When avalanche terrain is excluded, the next most similar neighbor is selected instead if an avalanche-prone location is identified during the nearest neighbor search.

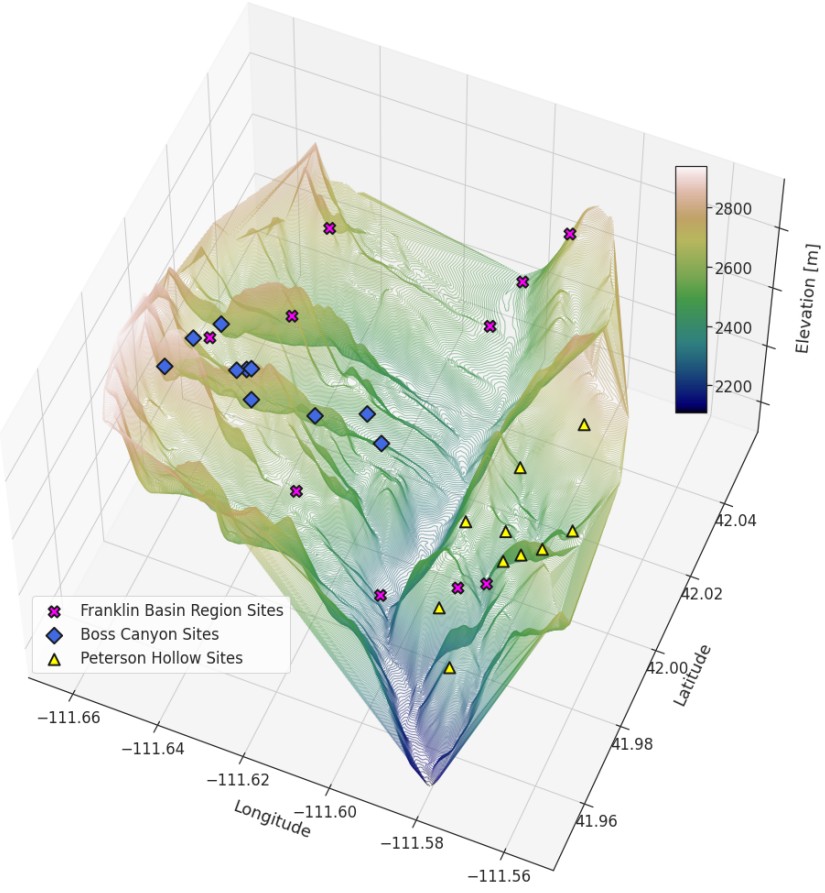

**Figure 5.** A 3-dimensional representation of the GMM output of the 10 most representative physiographic feature domain locations for Franklin Basin, Boss Canyon, and Peterson Hollow.

## 2.4 Snow Survey Protocol

The 10 sampling locations identified by the GMM were the focus of a snow depth survey to retrieve in-situ depth measurements. Simultaneously with the snow-on LiDAR flight on Mar 28, 2021, two researchers conducted a snow survey of the Hell's Kitchen Canyon study subbasin. The researchers reached the 10 locations at coordinates retrieved via a handheld Garmin InReach GPS device. The field team measured depth at each site throughout the subbasin via graduated snow depth probes. Each researcher took 4 measurements spaced evenly at the prescribed locations and recorded the average of the 8 measurements. The first measurement was taken at approximately 9:40 am with an air temperature of 5.8°C. Temperatures warmed throughout the day to 10.6°C at the time and location of the final measurement. Weather was clear during sampling, and the nearby Dream Lift - KUTGARDE14 weather station (elev: 2220 m, 41.97°N, 111.54°W) measured the latest precipitation event, five days prior, with 0.46 cm of rain and 0.15 cm of snow. The most recent significant precipitation event measured at the station occurred

twelve days prior, accumulating 2.59 cm of rain and 0.69 cm of snow. Temperatures over the past month at the station were consistently above 4.4°C during the day while remaining below freezing at night, indicating melt periods in the snowpack with few accumulation periods.

## 2.5 Snow Depth Regression Model

We model basin snow depth distribution throughout the study domain with the application of a Gaussian Process Regression
(GPR) model (Figure 6). Gaussian processes are a method of supervised ML that resolves a probability distribution (Gaussian) of multiple multivariate functions with joint Gaussian distributions to fit a dataset (Williams and Rasmussen, 2006). The probabilistic and non-parametric nature of a GPR allow it to effectively capture underlying patterns even with minimal training data. A GPR model may be expressed as

$$f(\mathbf{x}) \sim GP\left(m(\mathbf{x}), k\left(\mathbf{x}, \mathbf{x}'\right)\right), \tag{5}$$

where $\mathbf{x}$ represents a set of observations, $m$ represents the mean function, defined as $\mathbb{E}[f(\mathbf{x})]$, and $K(\mathbf{x}, \mathbf{x}')$, a covariance function for all possible pairs of data points for a given set of hyperparameters.

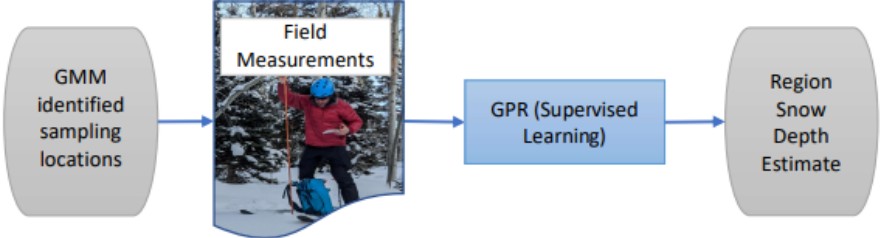

**Figure 6.** Workflow of the snow estimation model. The model relies on the GPR which collates static LiDAR features and point snow depth measurements to produce gridded snow depth estimates. In Hell's Kitchen Canyon, in-situ measured snow depths were used for the training data, while point observations of LiDAR-measured snow depth were used in the other domains.

The GPR predicts the dependent (target) variable snow depth for each point in the domain, given the independent variables (i.e. topographic parameters) and a prior covariance specified by a covariance function (kernel). GPR is categorized as a supervised ML regression technique, denoting that known values of the dependent variable are used during training to define a
225 covariance-based relationship with the independent variables and to predict at locations where the target variable is unknown. The applied kernel determines the shape of the posterior distribution of the GPR. We use a radial basis function (RBF) (also referred to as a Gaussian or Squared Exponential covariance function), $k$, defined by Eq. 6, where $x_i$, $x_j$ are two data points, $d$ is the Euclidean distance between the two points, and $l$ is a length-scale parameter.

$$k(x_i, x_j) = exp\left(\frac{-d(x_i, x_j)^2}{2l}\right) \tag{6}$$

The parameter $l$ controls the rate at which the correlation between two points decreases concerning distance and influences the "smoothness" of the prediction function. To determine a value for $l$, we perform a grid-search across a range of [0.01,1] for every point, with the selected value being that which maximizes the log-marginal likelihood.

    Within Hell's Kitchen Canyon, field-measured snow depths from 10 locations are used as training data to estimate basin-wide snow depth based on sites outside of the estimation domain. For the other model scenarios which lack snow survey data,

LiDAR-derived snow depths at the GMM-identified locations are used to train the GPR model. In this approach, individual LiDAR point measurements serve as synthetic snow probe samples. To prevent information leakage between training data and model validation, these point measurements are excluded from the validation dataset.

## 2.6   Model Evaluation

We measure the performance of the model as the difference in the predicted snow depth from the LiDAR-derived snow depth

throughout the basin. The standard metrics of mean bias error (MBE) (Eq. 7) and root mean squared error (RMSE) (Eq. 8) are used to track and compare performance across the various model scenarios. MBE and RMSE are defined here as,

$$MBE = \frac{1}{n}\sum_{i=1}^{n}(y_i - \hat{y_i}),\tag{7}$$

$$RMSE = \sqrt{\frac{1}{n}\sum_{i=1}^{n}(y_i - \hat{y_i})^2},\tag{8}$$

where, for both formulas, $n$ is the total number of points in the domain, $\hat{y_i}$ is the estimated snow depth and $y_i$ is the measured snow depth. Additionally, we report the Kling-Gupta efficiency (KGE) score of estimates (Eq. 9). KGE represents the goodness of fit between simulations to observations and incorporates the Pearson correlation coefficient ($r$), a term representing the variability of prediction error ($\alpha$), and a bias term ($\beta$). KGE ranges from -inf to 1.0, with larger values indicating greater simulation efficiency and a KGE of 1.0 indicating perfect reproduction of observations (Gupta et al., 2009; Knoben et al.,

2019).

$$KGE = 1 - \sqrt{[r-1]^2 + [\alpha-1]^2 + [\beta-1]^2}$$
$$\text{where: } r = \frac{cov(y_i),\hat{y_i}}{\sigma(y_i)*\sigma(\hat{y_i})}, \ \alpha = \frac{\sigma(\hat{y_i})}{\sigma(y_i)}, \ \beta = \frac{\mu(\hat{y_i})}{\mu(y_i)}\tag{9}$$

For further analysis of results, we define distribution similarity using the Kolmogorov-Smirnov (KS) test to define the KS statistic ($D$) defined by the equation,

$$D_n = \sup_x |F_n(x) - F(x)|\tag{10}$$

Where $F_n(x)$ is the empirical distribution function of the estimates, $F(x)$ is the cumulative distribution function of the reference distribution, and $\sup_x$ denotes the supremum or the maximum value of the absolute difference across all values of $x$. The

KS statistic measures the largest absolute difference between two distributions on a scale of 0 to 1, with a value of 0 indicating identical distributions (Conover, 1999).

To address the study objectives outlined in Section 1, we assess the model's performance across two distinct scenarios. First, we validate the model by estimating snow depth within a subbasin, utilizing only sampling sites from that basin. We compare the accuracy of snow depth estimates obtained from optimally located sites versus those from randomly selected sites. We also investigate how the number of sampling locations affects model accuracy by testing sparse measurements (e.g., 10 sites) against a more extensive dataset (e.g., 100 sites).

In the second scenario, we apply the validated model to estimate snow depth across the larger Franklin Basin, using sampling sites from selected subbasins (Objective ii). For this, we use either field-measured sample sites from Hell's Kitchen Canyon or point observations derived from LiDAR snow depth values as the training data for the GPR model. In both scenarios, we explore the impact of excluding avalanche-prone terrain from the sampling locations (Objective iii). Avalanche-prone cells are excluded during the GMM process and then reinstated for GPR snow depth estimation. The resulting snow depth estimates are then compared with those derived from the full-cell domain.

## 3 Results

### 3.1 Algorithm Validation

We validate the estimation capability of the GMM-GPR model within the study domain by processing snow depth estimates for the Boss Canyon and Peterson Hollow subbasins with lidar point observations sites in each catchment and comparing the scoring metrics (Table 3). In each tested scenario, optimally located sampling sites by the GMM resulted in reduced RMSE and improved MBE and KGE scores over randomly located sites. With 10 training sites optimally located by the GMM, the snow depth estimate results in RMSEs of 0.37 and 0.19 m for Boss Canyon and Peterson Hollow, compared to 0.43 and 0.33 m for random sites, respectively (Figure 7). The resulting estimate is only slightly improved when increased to 100 optimally located sites. Comparing sampling sites at random locations throughout the watershed rather than algorithm-identified, the model RMSE for 10 sites increases by 16% and 74%, with a greatly reduced bias for Boss Canyon and Peterson Hollow, respectively. Increasing the sampling to 100 randomly located sites slightly reduced the RMSE for Peterson Hollow compared to 10 random sites, with an improvement in MBE, while the error was exacerbated for Boss Canyon. The increased random sampling rate still results in greater error than just 10 optimized locations. For each scenario, modeled snow depth was slightly underestimated. The full sensitivity analysis may be found in Figure S1.

### 3.2 Impact of Avalanche Terrain Removal

The exclusion of avalanche-prone terrain had minimal influence on the resulting model estimation. When the high-risk cells are excluded from the potential sampling domain for Boss Canyon, the RMSE of the subbasin snow depth estimate for 10 training locations was increased by 2.7%. The MBE remains unaffected and the KGE is slightly reduced from 0.54 to 0.47. For 10 sites

**Table 3.** Model scenario and accuracy metrics for snow depth estimates of the Boss Canyon and Peterson Hollow subabsins and the full Franklin Basin region. Each scenario was executed for randomly selected and optimal locations (GMM) within the respective sampling subbasin, with 10 or 100 locations, and with and without avalanche-prone (Avy.) terrain. Model-identified training locations show improved estimation performance over randomly selected sites. Increasing the number of optimized sites provides slight improvement to accuracy in most scenarios

| Sampling Region | Number of Locations (Methodology) | | Including Avy. Terrain | | | Excluding Avy. Terrain | | |
|---|---|---|---|---|---|---|---|---|
| | | | RMSE (m) | MBE (m) | KGE | RMSE (m) | MBE (m) | KGE |
| Boss Canyon | 10 | (Random) | 0.43 | -0.13 | 0.36 | 0.43 | -0.13 | 0.36 |
| Boss Canyon | 100 | (Random) | 0.66 | -0.26 | 0.19 | 0.66 | -0.26 | 0.19 |
| Boss Canyon | 10 | (GMM) | 0.37 | -0.05 | 0.54 | 0.38 | -0.05 | 0.47 |
| Boss Canyon | 100 | (GMM) | 0.31 | -0.06 | 0.75 | 0.32 | -0.06 | 0.74 |
| Peterson Hollow | 10 | (Random) | 0.33 | -0.16 | 0.49 | 0.33 | -0.16 | 0.49 |
| Peterson Hollow | 100 | (Random) | 0.28 | -0.06 | 0.59 | 0.28 | -0.06 | 0.59 |
| Peterson Hollow | 10 | (GMM) | 0.19 | 0.0 | 0.77 | 0.21 | -0.03 | 0.71 |
| Peterson Hollow | 100 | (GMM) | 0.18 | -0.01 | 0.8 | 0.18 | -0.01 | 0.8 |
| Franklin Basin | 10 | (Random) | 0.43 | -0.08 | 0.53 | 0.43 | -0.08 | 0.53 |
| Franklin Basin | 100 | (Random) | 0.51 | -0.16 | 0.42 | 0.51 | -0.16 | 0.42 |
| Franklin Basin | 10 | (GMM) | 0.34 | -0.06 | 0.72 | 0.43 | -0.08 | 0.53 |
| Franklin Basin | 100 | (GMM) | 0.37 | -0.07 | 0.63 | 0.51 | -0.16 | 0.42 |

within Peterson Hollow, when accounting for avalanche terrain, the subbasin estimate RMSE expresses an increase of 11%, and a slight increase in MBE and reduction of KGE. The potential for optimal sampling sites being identified in avalanche-prone terrain increases as the number of sites increases, though the model remains robust when increasing the sampling site total and excluding the terrain. We found no significant change in the model score when the training set was increased to 100 sites outside of high-risk terrain. The Boss Canyon watershed estimate RMSE improves by 3.2% and KGE decreases by 1.3%. In Peterson Hollow, the accuracy metrics did not change with the increase in the number of sites considered.

### 3.3 Estimation Beyond the Sampling Region

Expanding the estimation domain beyond the smaller subbasin sampling spatial bounds results in effective snow depth modeling at the greater basin scale. We compare the results of the basin estimate for the various scenarios of 10 sites in a subbasin excluding avalanche terrain (Figure 8). For the Franklin Basin sampling instance, the RMSE of the basin estimate exhibits an RMSE of 0.34 m, MBE of -0.06 m and KGE of 0.72. All subbasin sampling domain estimations result in RMSEs similar to that of the basin sampled estimate (within 28%), with Boss Canyon providing the lowest RMSE (0.37 m) of the three subbasins and Hell's Kitchen Canyon the largest (0.45 m). Hell's Kitchen Canyon also provides the lowest KGE (0.39), while Peterson

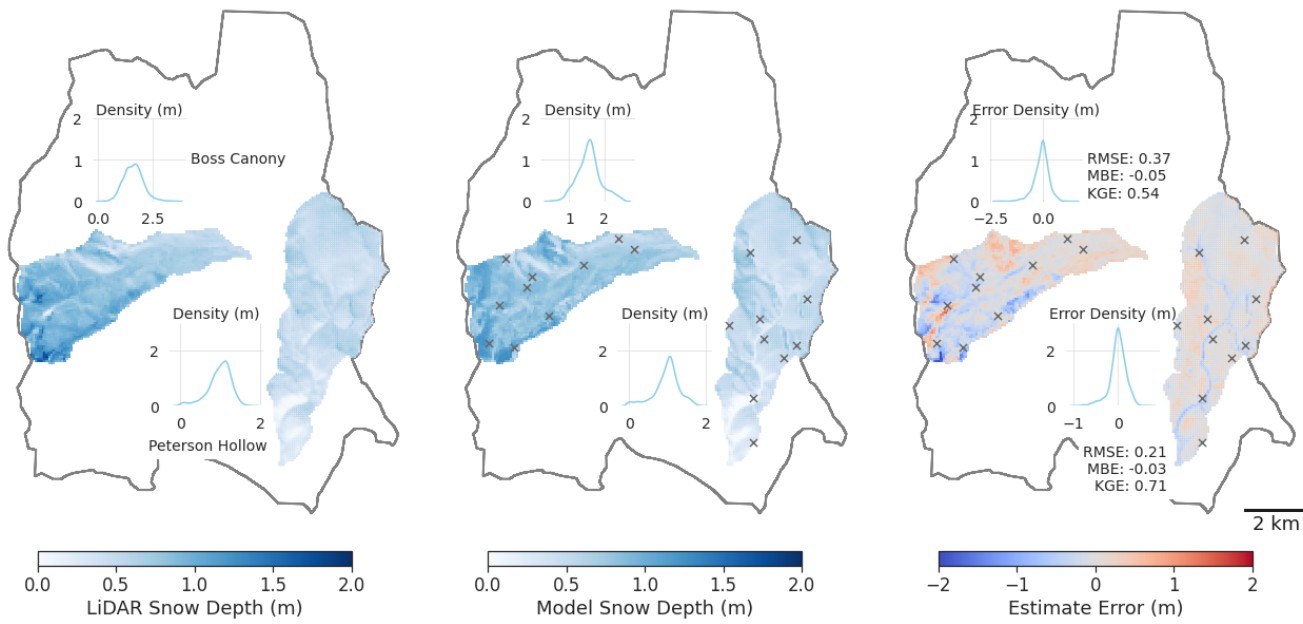

**Figure 7.** Subbasin Lidar snow depth, estimated snow depth, and estimation error for Boss Canyon and Peterson Hollow for 10 model-identified optimal sampling locations (x) and associated accuracy metrics.

Hollow maintains the highest (0.59). The absolute value of MBE is within 0.22 m for each estimate, with Boss Canyon and Hell's Kitchen Canyon slightly overestimating the basin's snow depth, while Peterson Hollow underestimates on average.

While the Hell's Kitchen Canyon derived estimate exhibits slightly greater RMSE than the Boss Canyon and Peterson Hollow estimates, the overall snow depth distribution is more similar to the true distribution (Figure 9). The Hell's Kitchen Canyon estimated snow depth exhibits a $D_{HK}$ of 0.13, and only the Franklin Basin derived estimate is more similar to the true distribution, with a $D_{FB}$=0.10. The overestimation of the Boss Canyon estimate can be observed in the distribution plot and with the greatest KS statistic of $D_{BC}$=0.28. For all subbasin estimates, errors increase with elevation (Figure 10).

When comparing the results of the estimates while excluding avalanche-prone terrain to estimates that included the terrain, we observe minimal performance loss in basin-scale estimation. The result of excluding avalanche terrain in the Franklin Basin sampling domain did not affect estimation error, as the 10 most characteristic sites are located outside of avalanche terrain. Excluding the terrain within Boss Canyon results in a 2.5% increase to RMSE and an improvement of MBE from 0.17 m to 0.15 m to basin estimates. While Peterson Hollow estimates exhibit an increase of 7.5% in RMSE and a 12% drop in MBE. No instance of included avalanche terrain was executed in the Hell's Kitchen Canyon analysis due to the constraint of having only one snow survey, which did not allow for sampling avalanche-prone areas.

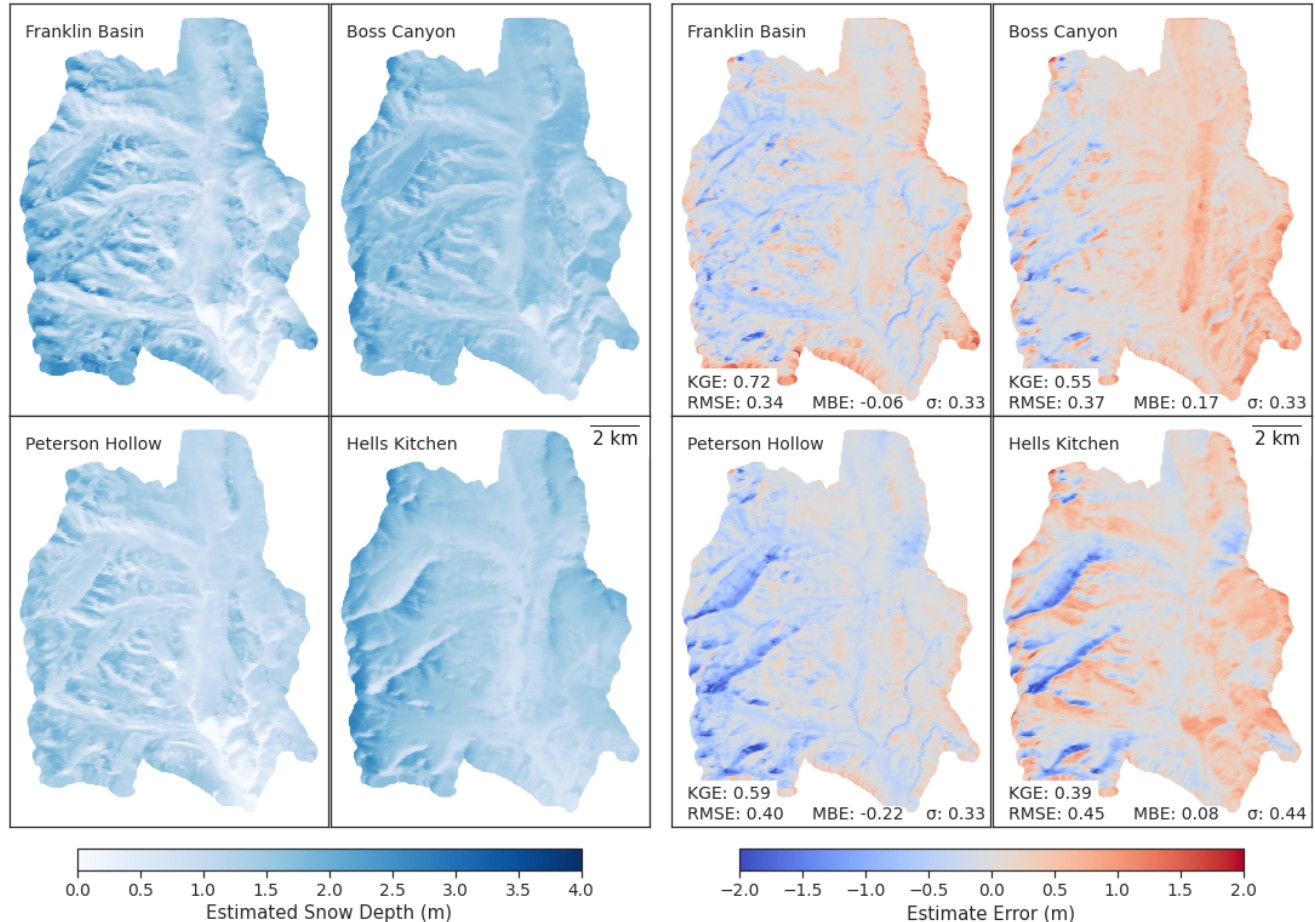

**Figure 8.** Basin-scale snow depth estimates (left), and snow depth estimate errors (right) with associated accuracy metrics derived from the sampling domains of: Franklin Basin (top left), Boss Canyon (top right), Peterson Hollow (bottom left), and Hell's Kitchen Canyon (bottom right). The estimates were derived with 10 GMM-identified, optimal sampling locations outside of avalanche-prone terrain to train the GPR model.

## 4 Discussion

### 4.1 Performance Across Subbasins

For each modeled scenario, the model consistently produced accurate snow depth estimates based on a small number of training locations and with better accuracy than random sampling (Table 3). Both the Boss Canyon and Peterson Hollow scenarios showed small decreases in accuracy when the estimation was expanded to the full basin, though they maintained similar MBEs and distributions. Peterson Hollow exhibits a greater drop in accuracy than Boss Canyon when estimating beyond its border due to its more homogenous feature distribution. Despite the decreased performance, the larger region-scale estimate from the

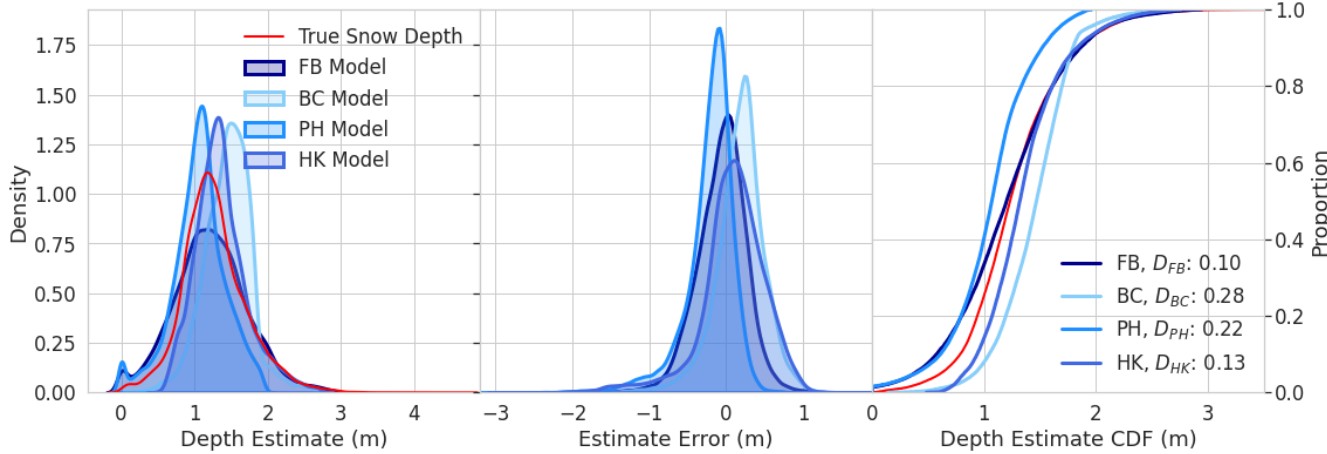

**Figure 9.** Distribution of basin-scale snow depth estimates for the four sampling domains, where FB indicates Franklin Basin, BC: Boss Canyon, PH: Peterson Hollow, and HK: Hell's Kitchen Canyon. *D* is the Kolmogorov-Smirnov statistic indicating distribution similarity between the estimate and true snow depth.

GMM-identified Peterson Hollow sites exceeds the accuracy of random sites throughout the basin. In Hell's Kitchen Canyon, a subbasin outside of the Franklin Basin domain, the similarity of the basin-scale estimate distribution exemplifies it can adequately estimate snowpack in the basin, though with slightly worse scoring metrics than the other subbasins. This indicates that a smaller sampling catchment with optimized sampling sites can be accurately applied to model a larger or similar basin without sacrificing performance, though perhaps up to a limit.

## 4.2 Sampling Constraints and Practical Considerations

Small improvements in model performance were observed as we increased the number of sampling locations beyond 10 sites and when considering a larger spatial sampling domain. However, increasing the number or spatial range of sites makes it infeasible for a group of samplers to collect snow depth samples in a single day. The ability to collect data quickly is critical to accurately representing the snowpack before changes occur, such as melting or new accumulation. Additionally, access to sampling locations and the availability of citizen-scientist-collected data may be limited to certain areas. For example, Hell's Kitchen Canyon is a popular outdoor recreation area with nearby parking and trail access. It is more likely to see traffic, which may result in more data collection than in a more remote area. Under different conditions, a sampling team may reach more than 10 sites in a day, such as with multiple teams to sample simultaneously, or there may be many snow probe data points at a higher density in a larger region. Conversely, sampling and the quality of sampling may be more challenging in a more remote or complex environment.

While 10 sites were selected as the focus of this study, we did not aim to determine an optimal number of locations for a snow survey. During model validation, we observed variation in performance from incremental increases in the number of training sites (e.g., increasing from 10 to 11 sites). However, the overall trend across all basins was consistent: increasing the number

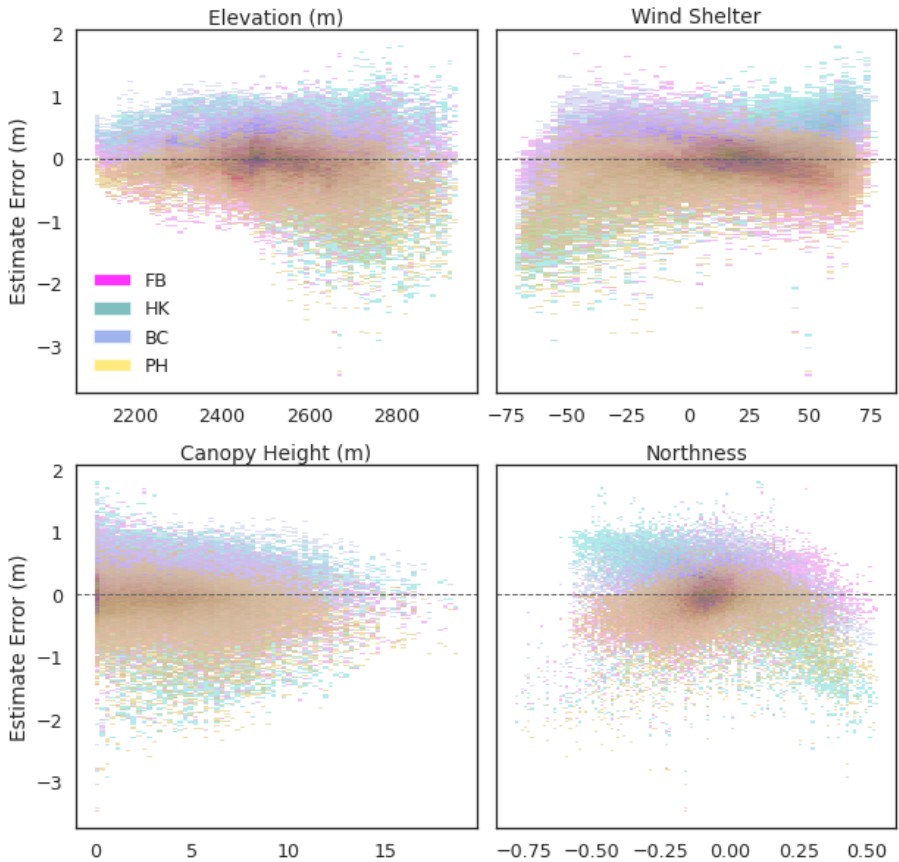

**Figure 10.** Distribution of basin-scale snow depth estimate errors for the individual physiographic features. Estimate variance is greater, particularly at higher elevations and at the extremes of the wind shelter spectrum. The Hell's Kitchen Canyon estimate exhibits the largest variance across all variables.

of points did not significantly improve model performance over a smaller sample, such as 10 (Figure S1). We hypothesize that an optimal sample number is dependent on factors such as basin size, regional characteristics, and terrain complexity. Therefore, determining the optimal or projected number of physically collected sampling points becomes a complex calculation of sampling area, group size, region traffic, and community engagement with data collection. Future work shall identify the lower
bounds of sampling sites and region size that may still produce representative results. For the subbasins in our study and a group of two samplers, approximately ten sites were the most they could safely measure in one day without mechanized travel, and we consider the ten site simulations relevant and realistic for similar environments and scales. Additionally, future work could explore a single-site (e.g., SNOTEL-based) framework to assess temporally continuous model performance. However, due to a lack of validation data, this was beyond the scope of our study. We anticipate that using only one or very few sampling sites
would lead to a highly biased and overfitted model, as the GPR would struggle to capture the underlying spatial relationships.

It should be noted that the specified accuracy range of the LiDAR sensor (<0.03-0.25 m) may be significant, considering the depth of snowpack in the region, and warrants further examination. For the Hell's Kitchen Canyon simulation which relies on the field measurements, errors in LiDAR data or in the depth probe measurements may lead to biases in the overall snow depth estimate. Although the impact of uncertainty on the study's results is unclear, the synthetic sampling method in the other domains mitigates additional error by directly comparing LiDAR measurements to LiDAR measurements. Further research should involve validating the model with diverse LiDAR sources and additional ground-truthed datasets to quantify and characterize the error.

## 4.3 Influence of Avalanche-Prone Terrain

The model showed low sensitivity to the consideration of avalanche-prone terrain. In all scenarios, excluding high-risk terrain led to minimal or negligible increases in estimation RMSE for the tested regions. Thus, snow samplers' safety is protected without sacrificing significant estimate accuracy, as they do not need to physically sample in higher risk terrain. While the absence of terrain does not greatly impact the average basin estimate, it does have a larger effect on the estimation bias observed. The Boss Canyon estimate suffers from more underestimation of snow depth, likely due to the underrepresentation of higher elevations, which are largely associated with avalanche-prone slopes within the canyon. Additionally, only 5% of the cells within Franklin Basin and 10% within Hell's Kitchen Canyon consist of terrain defined as avalanche-prone. This allows for an adequate sampling domain outside of avalanche risk, with a broad enough range of features to represent the region in safe-to-sample locations.

While a slope angle of 30° is a commonly referenced threshold for avalanches, accurately determining avalanche terrain is often more complex. Avalanches may run out to flatter areas beyond the risk terrain analyzed in this study or may propagate horizontally to areas of slope angles below the threshold. In steeper, more complex terrain, the area unavailable to safely sample may be much larger, and the estimation performance observed in Franklin Basin may not be exhibited in other regions. Future work shall aim to apply the model to additional regions of steep terrain to determine the potential limits of domain exclusion from avalanche terrain.

## 4.4 Sampling Domain Similarity and Model Performance

A key assumption of the methodology is that there is enough characteristic variety in the sampling domain to represent the variability in the feature space of the estimation region. We can observe this dependency on sampling and estimation domain similarity by comparing the physiographic features of the various domains (Figure 2). Hell's Kitchen Canyon's canopy height distribution appears very dissimilar to the other canopy distributions. This is due to the spatial resolution difference between Hell's Kitchen Canyon and the other domains. Hell's Kitchen Canyon maintains a higher resolution which is capable of capturing the smaller variations in canopy height, whereas this detail is averaged out during the upscaling process of the other regions. The increased variance of values likely contributes to Hell's Kitchen Canyon model resulting in poorer performance than the other models, however, we posit that with uniform resolution, the feature space would present more analogous to Franklin Basin's and thus improve results.

By comparing the individual topographic features to estimate error, we observe the largest variance of error correlated with higher elevations and at the extremes of the wind shelter metric (i.e., fully exposed or sheltered terrain) (Figure 10). In a small, snow-fed catchment (<1 $km^2$), wind redistribution of snow may be the most important factor for snowpack accumulation and persistence, with the best predictor of single-point snow depth to be its elevation relative to the neighboring terrain at a 40 m radius (Anderton et al., 2004). In practice, however, obtaining or generating accurate wind scales for snow depth estimation at the basin scale is challenging and requires the downscaling of computational fluid dynamic models or high-resolution numerical weather prediction models (Reynolds et al., 2021). Another approach is to classify persistent locations of wind-drifted snow from remote sensing imagery for feature development, though requires historical analysis and the addition of multiple data sources. Overall, a more detailed investigation of the local wind dynamics in a sampling region beyond the simplified wind shelter metric applied in this study may be considered to improve model performance.

While redistribution factors such as wind play a major role in snow depth distribution, the elevation relationship is also central. In alpine environments, snow depth tends to increase with elevation up to a point correlating with prominent rock coverage and then decreases beyond. The decrease in depth at high elevation is likely due to redistribution and preferential deposition factors, such as wind transport, avalanching, and sloughing from steeper to shallower slopes (Grünewald et al., 2014). Additionally, orographic precipitation dynamics can result in varying elevation precipitation patterns (Roe and Baker, 2006). We can observe the model elevation dependence, particularly in the Boss Canyon results, which contain the highest percentage of high-elevation cells among all subbasins. This leads to the most accurate depth estimates at higher elevations and ridgelines in the large-scale analysis. In contrast, the lower subbasins of Peterson Hollow and Hell's Kitchen Canyon show weaker performance in estimating these areas.

More normally distributed in feature space than the other regions, Peterson Hollow lacks the topographic diversity to accurately estimate snow depth in the highest elevations and particularly on northerly aspects. The lack of high elevations in Peterson Hollow likely fails relative to the other subbasins in representing the high-elevation transport dynamics of the larger Franklin Basin, resulting in high elevation underestimation. Alternatively, Boss Canyon encompasses a broader feature range, more similar to the greater basin, and results in a more accurate estimate. Thus, selecting a diverse and representative sampling area is advantageous when modeling beyond the spatial bounds of the sampling area.

The distinction in basin elevation distribution between the regions is identifiable. Hell's Kitchen Canyon, which sits at a lower elevation than the overall Franklin Basin, exhibited greater error at high elevations and along mountain ridgelines. and Boss Canyon contains the most high-elevation terrain (Figure 8). In contrast, Boss Canyon, with a higher average elevation, showed greater snow depth estimation errors at lower elevations. These results suggest that elevation plays a crucial role in model accuracy, and selecting a representative sampling domain is key to minimizing estimation bias.

## 4.5 Implications for Model Transferability and Future Work

To improve model transferability, incorporating additional features such as radiative forcing, land cover, and atmospheric properties may help compensate for the limitations of the pyshiographic feature space. In this study, we constrained the feature space to maintain efficient domain preprocessing and execution using a single LiDAR dataset. However, expanding the fea-

ture space could enhance model performance, particularly in regions with more complex snowpack dynamics. Future studies should explore the benefits of integrating multi-source datasets to refine snow depth estimations across diverse topographic environments.

While this methodology was shown to be effective in the study region of Northern Utah and Southern Idaho, application in other regions is necessary to evaluate performance across various topographic environments and snow climates. The snow distribution of Franklin Basin at the time of sampling presented as near-Gaussian (Figure 3). The expectation of a Gaussian distributed target variable is a key assumption of a GPR model. While snow depth often appears as Gaussian in subalpine and non-ephemeral regions, non-Gaussian distributions are observed in more complex or ephemeral environments (He et al., 2019; Ohara et al., 2024). Application of the model in more diverse snow environments, including basins with greater exposure to wind-swept terrain above treeline, and basins with snow-free areas is needed to understand transferability to regions less likely to meet the Gaussian assumption. Additionally, this study is performed for a single date towards the end of the snow season. More sampling dates and at periods throughout the snow season, especially after snow accumulation events, should be investigated to determine the temporal strength and sensitivity of the model.

## 5 Conclusions

The development of low-cost, near-real-time snow estimation is critical for water resource monitoring, particularly in remote, unmonitored regions. With this study, we introduce a methodology leveraging physiographic features derived from one-time captured snow-off LiDAR and a small number of in-situ sampling points to generate region-scale snow depth estimates with low-cost and high temporal efficiency. A two-step ML workflow is applied. First, a Gaussian mixture model is used to locate optimal sampling locations based on feature representation. These locations are then sampled for point snow depth values and used to train a Gaussian process regression algorithm to estimate broad-scale snow depths. We find that with few (i.e., 10) optimized sampling points, the model is effective at estimation both at the subbasin and greater basin scales. The solution proves robust for model scenarios encompassing both the sampling subbasin (Boss Canyon and Peterson Hollow) and for a sampling subbasin (Hell's Kitchen Canyon) outside of the estimate bounds. With the goal of lowering the avalanche risk of individuals sampling snow depths in the field, we test the sensitivity of the model to the exclusion of avalanche-prone terrain from the sampling domain. When high-risk terrain is removed, we observe the model produces snow depth estimates with minimal performance loss. Results demonstrate that a relatively small number of optimal sampling locations can effectively model snow depth across a broader region, reducing the need for extensive sampling campaigns. This approach provides a use case for accurate snow modeling via application of low-cost snow probe measurement while prioritizing safety and systematically reducing personnel danger in the field.

*Data availability.* The data that support the findings of this study are openly available from the data hosting website Hydroshare.org, from the repository at https://doi.org/10.4211/hs.34ce22e4df10463cb053bb63d19d6672

*Author contributions.* **D. Liljestrand**: Conceptualization, Data curation & Field sampling, Formal analysis, Methodology, Software and code development, Validation, Visualization, Writing – original draft, Writing – review & editing. **R. Johnson**: Writing – review & editing. **B. Neilson**: Funding acquisition, Project administration, Supervision, Writing – review & editing. **P. Strong**: Field Sampling, Writing – review & editing. **E. Cotter**: Code development.

*Competing interests.* The authors declare that they have no conflict of interest.

*Acknowledgements.* This work was made possible through funding provided by the United States Geological Survey, the Logan River Observatory (USU), and the Cooperative Institute for Research to Operations in Hydrology (CIROH) through the NOAA Cooperative Agreement with The University of Alabama (NA22NWS4320003). The authors would like to acknowledge the contribution of the following individuals and organizations: Carlos Oroza PhD (UofU), Michael Jarzin Jr. (UofU), Data hosting and mobile collection provided by Community Snow Observations (communitysnowobs.org), LiDAR flown by Aero-Graphics Inc., SLC, UT, field safety training and planning provided by American Avalanche Institute, Jackson, WY.

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
