# Peer review of "Leveraging Snow Probe Data, LiDAR, and Machine Learning for Snow Depth Estimation in Complex Terrain Environments"

_EGUsphere, 2024_

## Referee Comment (RC2)

Liljestrand et al. employ a two-step machine learning framework to derive basin-wide, high-resolution snow depth from a limited number of in-situ snow depth samples. The framework identifies optimal sampling locations based on areas with representative physiographic features, then applies a Gaussian Process Regression model to estimate snow depth at other locations based on the physiography at a pixel. This work presents a compelling method to estimate basin wide snow depth based on a limited number of user collected samples. In this document I outline some minor revisions to improve the manuscript prior to publication.

**General comments:**
- The authors emphasize that the model relies on in-situ samples and static terrain features (lines 85-86). This overlooks the role of snow-on lidar in the snow depth estimation process. The GPR is a supervised ML model which relies on the snow-on lidar data for model training. Model results are then validated on the same day as the lidar flight, meaning that the model had access to proximal snow depth data from the same day in its estimations. A key question that is not addressed is how transferable the GPR model is to other times. If citizen scientists collect data in different years or periods within the snow season, how would this impact model results? I understand it would be difficult to address this question given lidar data availability in the study area, but this seems worthy of discussion since it impacts the applicability of the methods for their intended use.
- The introduction discusses the importance of snowpack as a water source, but the manuscript focuses on snow depth with little mention of SWE. I recommend adding a paragraph regarding the decision to focus on snow depth and the potential future applicability to SWE.
- The authors primarily use 10 as the number of optimally placed sampling locations. At one point using five samples is mentioned, but I do not see results for this. To me, this is a key question of the study: how few locations can be sampled while still getting quality results? Additionally, Snotel represent a 'one sample' framework. How much advantage does multiple locations pose over a single sample? This is important since Snotel provide the advantage of temporally continuous data. I see this is mentioned in the discussion as a subject of future work. Based on the available data it would be feasible to reduce the number of sampling sites and produce results. If this is out of the scope of this paper, some justification could be provided.
- Table 2 is the only place where results are presented for more than 10 samples. If results were calculated iteratively up to 100, it could be helpful to visualize errors with the number of samples on a line plot, even if just added to the supplement.
- The figures only visualize model error. It would be helpful to include figures which visualize both lidar snow depth and model snow side-by-side (addressed in line-by-line comments).
- The Discussion section could benefit from subheadings to improve readability.

**Substantive line-by-line comments:**

Title: The title broadcasts the "leveraging of citizen science data" but no such data appear to be utilized in this study. I understand that the approach/findings of the paper have implications for

guiding citizen science data collection, but I feel the title has potential to be misleading. Consider reframing.

Line 2. 'Address gaps in basin-scale snowpack modeling' is a bit vague. Can you briefly describe the gap you are addressing? Increasing spatial information of the snowpack?

Line 9. Maybe state which dataset represents the "true snow depth distribution".

Line 12. Add "in the training data" after "excluded".

Line 34. Meromy et al., 2013 and Herbert et al., 2024 (references at end of document) are more recent papers which explore Snotel representativeness.

Line 37-39, 144. Is the assumption of normally distributed snow depth key to the methodology? As in, does the GPR make assumptions about the distribution of snow depth when making predictions? If yes, this assumption could be explored further in the discussion. Does the methodology deteriorate if snow depth is not normally distributed? If this is not key to the methodology, this information feels unnecessary. Additionally, it would be useful to show the lidar snow depth histogram to convince readers it follows a normal distribution.

Line 69: Well, not "anytime" (weather/clouds can still be a factor in the snow-free season. Consider rephrasing.

Line 70. I was a bit confused here when you mention 'snow-free lidar data'. If my understanding is correct, saying something like: 'physiographic data from snow-free lidar scans' would make this sentence easier to follow.

Line 74 (paragraph). Citing examples of papers which use citizen science could be beneficial here. See Crumley et al. 2021.

Figure 7. I recommend adding panels which show the modeled snow depth and lidar snow depth (in addition to the delta snow depth). This allows for the reader to make an easier visual comparison of the two maps. I also recommend adding sampling locations to the map.

Figure 7. Is there an explanation for the horizontal blocks of similar error? There appear to be blocks of ~5 horizontal pixels that tend to register the same error (and maybe the same snow depth?). Is this an artifact of the GPR? Topography?

Line 132: How was it decided that northerly is used as the upwind barrier direction? Is this the dominant wind direction in the area?

Figures 2, 10, etc.: The use of red (FB) and green (HK) lines/markers may render some figure unreadable those with red-green color vision issues. Please consider revising to make the figures more accessible.

Line 142: Please provide the date of the sampling effort. I don't see it anywhere in the document.

Lines 142-148: Was ground-truthing conducted for the lidar-derived snow depths? If so, what differences/errors were found?

Line 147-148. The explanation for why one study area wasn't upscaled could be clarified.

Figure 3 and Lines 134-138: One could argue that the avalanche runout zones (flatter slopes below the avalanche slopes) should be included as "avalanche-prone terrain", since the goal is to avoid measurements in dangerous zones.

Line 175-177. I don't see any results for the model which used five sampling locations.

Line 201. Any justification for the use of the GPR model? Pros/cons versus other models? Or just following the Oroza methodology?

Figures 7 & 8: It would be useful and interesting to show the maps of lidar snow depth and modeled snow depth in addition to the map of estimate errors (what is currently shown).

Figure 9. The left two plots are difficult to interpret based on the current colors. On the left I could barely find the curve with the lowest peak and in the middle plot I can only see four curves. Consider changing colors, using different line styles, or removing the fill on the curves.

Line 309: "streamflow forecasting" – it is odd that this is only mentioned in the conclusions but not earlier in the paper. Consider removing or providing more context earlier.

Line 310 (paragraph). In a similar vein to the avalanche terrain exclusions, I wonder how many appropriate sampling locations could be found. You currently choose the 'best' sampling location, but what if you selected the top 5 for each cluster? Then the samplers could select the location which is easiest to access. Maybe out of scope here, but just a thought!

Line 401. 'significant' has statistical implications. Maybe something like 'minimal losses' instead.

**Formatting and wording comments**

Lines 12, 29, and elsewhere: What does 'seamless' mean in these sentences?

Line 29: 'Products to produce' is a bit awkward.

Line 35: should be "snowpack" (no hyphen).

Line 37: Should be "snow depth" (no capital S)

Line 62: 'Aerial flown' is awkward. Maybe just 'aerial'.

Line 76. 'collected by such users via a mobile app platform'. Maybe 'reported' would be more appropriate.

Line 105: Careful with capitalization of cardinal directions here and in the rest of the document. No need to capitalize East, Easterly, etc.

Line 115: 'Snow-free' shouldn't be capitalized.

**References:**

Crumley, R. L., Hill, D. F., Wikstrom Jones, K., Wolken, G. J., Arendt, A. A., Aragon, C. M., et al. (2021). Assimilation of citizen science data in snowpack modeling using a new snow data set: Community Snow Observations. *Hydrology and Earth System Sciences*, *25*(9), 4651–4680. https://doi.org/10.5194/hess-25-4651-2021

Herbert, J. N., Raleigh, M. S., & Small, E. E. (2024). Reanalyzing the spatial representativeness of snow depth at automated monitoring stations using airborne lidar data. *The Cryosphere*, *18*(8), 3495-3512. https://doi.org/10.5194/tc-18-3495-2024

Meromy, L., Molotch, N. P., Link, T. E., Fassnacht, S. R., & Rice, R. (2013). Subgrid variability of snow water equivalent at operational snow stations in the western USA. *Hydrological Processes*, *27*(17), 2383-2400. https://doi.org/10.1002/hyp.9355

---

## Author Response (AR1)

**Manuscript Number:** Preprint egusphere-2024-3545
**Title:** Leveraging Citizen Science, LiDAR, and Machine Learning for Snow Depth Estimation in Complex Terrain Environments
**Subject:** Response to Manuscript Review Comments

Thank you for your thoughtful and constructive feedback on our manuscript. We appreciate the time and effort invested in reviewing our work and have implemented several revisions to enhance the clarity, transparency, and accuracy. We hope the revisions adequately address the comments and remain open to further suggestions or questions.

**Response to Reviewer #1:**

*The study tried to characterize and model the snow distribution using the Gaussian Process Regression (GPR) method and the Gaussian-based machine learning model (GMM). GMM and GPR method application for the Lidar observed snow distribution seems novel although they were tested using only one snapshot snow distribution. The land surface characterization in Figure 2 is nice, and the finding of elevation variability requirement for this method is interesting. However, since the transferability of this method may still be arguable, I recommend "major revision" for this review cycle for clarifications and further possible improvements. I have a few major points listed below:*

> *Comment 1:*
>
> The presentation of the data used in this study should be improved. The observed snow distribution by the airborne Lidar may be visualized and presented somewhere in the manuscript, perhaps instead of Figure 3. It will be informative for readers to see the variability and the extent of the dataset.
>
> **Response to comment 1:**
>
> **Thank you for your comment. Figure 3 (shown below) has been replaced with a map of the snow-on Lidar, as well as the distribution. Figures 7 and 8 (below) have also been updated to show the lidar snow depth and depth estimate maps along with the error maps. Note the scale of some figures have been shrunk to fit in this document.**
>
> **New Figure 3:**

[Figure]

**Figure 3.** Map and corresponding density distribution of the LiDAR-measured snow depth in Franklin Basin. The violin distribution of the snow depth is overlayed above a Gaussian distribution of 20000 randomly generated samples to express the near-normality of the snow depth distribution. Quartiles of the snow depth are shown as bold hashed lines, and the light grey hashed lines represent the Gaussian distribution quartiles.

**New Figure 7:**

[Figure]

**Figure 7.** Subbasin Lidar snow depth, estimated snow depth, and estimation error for Boss Canyon and Peterson Hollow for 10 model-identified optimal sampling locations (x) and associated accuracy metrics.

**New Figure 8:**

[Figure]

**Figure 8.** Basin-scale snow depth estimates (left), and snow depth estimate errors (right) with associated accuracy metrics derived from the sampling domains of: Franklin Basin (top left), Boss Canyon (top right), Peterson Hollow (bottom left), and Hell's Kitchen Canyon (bottom right). The estimates were derived with 10 GMM-identified, optimal sampling locations outside of avalanche-prone terrain to train the GPR model.

*Comment 2:*

Also, it is unclear when the LiDAR data collected. The snow distributions are highly dependent on season and year. From the snow distributions (Figure 9), I speculate that it must be late spring. Moreover, observation dates of the in-situ snow depth survey must be presented as well. Were they exactly same day? How good were they? Were they (field data vs. Lidar) consistent each other? It is unclear how the authors use actual field measured data. I would suggest adding a data list table.

**Response to comment 2:**

**We apologize for the confusion of the data metadata. The snow-on LiDAR and snow surveys were all performed on the same day in late spring (Mar 28, 2021). Text has been added to Sections 2.2 and 2.4 to address this, as well as a table provided of the data used in the study.**

**The field measured snow depths were used as the training data for the Hell's Kitchen Canyon subbasin model. This was the only basin where we performed sampling, and it was not covered by the lidar flight pattern. The samplers followed a rigorous protocol to reduce errors in measurements, however because the lidar data does not cover the sampling locations, we are unable to directly compare field samples to lidar. Text has been added to section 2.4 to clarify**

how the field samples were used, and to the Discussion section addresses the potential for error and uncertainty from the lidar data.

*Comment 3:*

The assumption for the methodology must be further clarified. Based on my understanding, Gaussianity in local snow distribution is required while it may not be true. I recall a recent publication in the same journal (TC) discussing non-Gaussianity of snow distribution (https://tc.copernicus.org/articles/18/5139/2024/). Assumption of local Gaussianity may be addressed in the limitation statement in the discussion as a reminder.

**Response to comment 3:**

**Thank you for the comment and the article recommendation. The GPR does rely on the target variable (snow depth) to be of a Gaussian distribution, and can lose robustness particularly for large outliers or heavy skewness. We justify the use of GPR in our study area based on the near normality of the LiDAR snow depth. We have added a figure of the LiDAR snow depth distribution (Figure 3) to Section 2.2 and provided statistical justification for the assumption. Text has been added to the Discussion section to expand on the Gaussian assumption, and to explain this methodology may suffer when snow depth does not follow this assumption.**

*Comment 4:*

I understand that there was no improvement by increasing sample number from 10 to100. It would be more useful if the author could quantify the ideal snow data point density (for instance, # of data point per unit area, perhaps). I understand it may be beyond scope of this study while lacking statement on potential transferability made this work just a case study based on single instantaneous snow distribution, which is rather weak.

**Response to comment 4:**

**Thank you for the comment. We agree the optimal number of sites is an interesting question, however we see it as different from the focus of this study. We aim to show that a small number (reasonable to be sampled in one day) of samples can effectively model a broader region and a large sampling campaign or dataset is not required. This is why we highlight only the 10 vs 100 site results (Table 2).**

**We have compiled results of the site sensitivity analysis in the supplemental materials (Figure S1).**

**We hypothesize the optimal number of sites would be highly dependent on basin size, region, terrain etc. and we did not have the snow LiDAR availability to test a large number of basins. We saw that there is large variation in the individual point-to-point increase of a small training dataset in the 4 basins analyzed, but the trend across all basins is similar; that a large number of points does not greatly improve performance. This can be seen in Figure S1.**

**We have included more text on this in the discussion, however we chose to provide the analysis results as supplemental materials as not to shift the focus of the study.**

**Supplemental Figure S1:**

[Figure]

*Comment 5:*

It is good to define the variables in the equations (2 through 4) as physical quantities (e.g. x = snow depth). Capital sigma (=covariance?) may be avoided because you use "summation" as same symbol.

**Response to comment 5:**

Per the suggestion, we have added the physical property to the variable description, as well as using lower case sigma in equations 2-4 instead of capital sigma to avoid confusion.

**Manuscript Number:** Preprint egusphere-2024-3545
**Title:** Leveraging Citizen Science, LiDAR, and Machine Learning for Snow Depth Estimation in Complex Terrain Environments
**Subject:** Response to Manuscript Review Comments

Thank for your thoughtful and constructive feedback on our manuscript. We appreciate the time and effort invested in reviewing our work and have implemented several revisions to enhance the clarity, transparency, and accuracy. We hope the revisions adequately address the comments and remain open to further suggestions or questions.

**Response to Reviewer #2:**

*Liljestrand et al. employ a two-step machine learning framework to derive basin-wide, high- resolution snow depth from a limited number of in-situ snow depth samples. The framework identifies optimal sampling locations based on areas with representative physiographic features, then applies a Gaussian Process Regression model to estimate snow depth at other locations based on the physiography at a pixel. This work presents a compelling method to estimate basin wide snow depth based on a limited number of user collected samples. In this document I outline some minor revisions to improve the manuscript prior to publication.*

**General comments:**
  *Comment 1:*

  The authors emphasize that the model relies on in-situ samples and static terrain features (lines 85-86). This overlooks the role of snow-on lidar in the snow depth estimation process. The GPR is a supervised ML model which relies on the snow-on lidar data for model training. Model results are then validated on the same day as the lidar flight, meaning that the model had access to proximal snow depth data from the same day in its estimations. A key question that is not addressed is how transferable the GPR model is to other times. If citizen scientists collect data in different years or periods within the snow season, how would this impact model results? I understand it would be difficult to address this question given lidar data availability in the study area, but this seems worthy of discussion since it impacts the applicability of the methods for their intended use.

  **Response to comment 1:**

  **Thank you for the comment, I will attempt to clarify our use of the snow-on LiDAR. In the study we do rely on individual (i.e. 10 per sub-catchment) grid cell snow depth LiDAR measurements for training. This was to supplement and serve as "synthetic" snow probe sample measurements in the locations we did not have the opportunity to field sample. The exception to this is in Hell's Kitchen Canyon, where we performed field sampling, so no snow-on LiDAR data was used in the training of that subbasin model at all. This allowed us to run the model for the multiple sub-catchments with the assumption that a point measurement from the lidar is a suitable replacement datapoint for using scare point snow measurements as training data. These snow-on LiDAR cells were then excluded from the dataframe for validation, so there was no information overlap in training and validation. Text has been added to Section 2.5 to clarify this methodology.**

**The question of temporal transferability is a key subject for future work. It was not in the scope to perform multiple LiDAR flights and snow surveys throughout the course of a snow season or across multiple seasons. This is a main interest point of future work to determine the temporal limitations of the framework. Text has been included in the Discussion Section to highlight this.**

*Comment 2:*

The introduction discusses the importance of snowpack as a water source, but the manuscript focuses on snow depth with little mention of SWE. I recommend adding a paragraph regarding the decision to focus on snow depth and the potential future applicability to SWE.

**Response to comment 2:**

**We have added a paragraph to the introduction to highlight the relationship between snow depth and SWE.**

*Comment 3:*

The authors primarily use 10 as the number of optimally placed sampling locations. At one point using five samples is mentioned, but I do not see results for this. To me, this is a key question of the study: how few locations can be sampled while still getting quality results? Additionally, Snotel represent a 'one sample' framework. How much advantage does multiple locations pose over a single sample? This is important since Snotel provide the advantage of temporally continuous data. I see this is mentioned in the discussion as a subject of future work. Based on the available data it would be feasible to reduce the number of sampling sites and produce results. If this is out of the scope of this paper, some justification could be provided.

**Response to comment 3:**

**Thank you for the comment. We have included results of the site number sensitivity analysis in the supplement materials (Figure S1) (Shown Below).**

**We see the question of the optimal number of sites as different from the focus of this study. We aim to show that a small number (reasonable to be sampled in one day) of samples can effectively model a broader region and a large sampling campaign is not required. This is why we focus on the 10 vs 100 site results (Table 2).**

**The optimal number of sites is an interesting question, however we hypothesize this number would be highly dependent on basin size, region, terrain etc. and we did not have the snow LiDAR availability to test a large number of basins. We saw that there is large variation in the individual point-to-point increase of a small training dataset in the basins analyzed, but the trend across all basins is similar; that a large number of points does not greatly improve performance. This can be seen in Figure S1.**

**The Snotel, one point framework, is additional future work we would like to perform**

particularly to explore the temporally continuous performance. However we did not have the validation data to perform this. Additionally we anticipate a one point, or extremely small sample size would result in a heavily biased and overfit model, as the GP would struggle to determine the underlying relationships of features.

We have included more text in the discussion section to  support this approach.

**Supplemental Materials Figure S1:**

[Figure]

*Comment 4:*

Table 2 is the only place where results are presented for more than 10 samples. If results were calculated iteratively up to 100, it could be helpful to visualize errors with the number of

samples on a line plot, even if just added to the supplement.

**Response to comment 4:**

**We have included results of the site number sensitivity analysis in the supplement materials (Figure S1).**

*Comment 5:*

The figures only visualize model error. It would be helpful to include figures which visualize both lidar snow depth and model snow side-by-side (addressed in line-by-line comments).

**Response to comment 5:**

**We have adjusted Figure 7 (Shown below) to display the lidar snow depth, and estimated snow depth Figure 8 (below) to display side-by-side the estimated snow depth for each subbasin result, as well as the respective errors.**

**Note that the scale of figures have been shrunk to fit in this document.**

**New Figure 7:**

[Figure]

**Figure 7.** Subbasin Lidar snow depth, estimated snow depth, and estimation error for Boss Canyon and Peterson Hollow for 10 model-identified optimal sampling locations (x) and associated accuracy metrics.

**New Figure 8:**

[Figure]

**Figure 8.** Basin-scale snow depth estimates (left), and snow depth estimate errors (right) with associated accuracy metrics derived from the sampling domains of: Franklin Basin (top left), Boss Canyon (top right), Peterson Hollow (bottom left), and Hell's Kitchen Canyon (bottom right). The estimates were derived with 10 GMM-identified, optimal sampling locations outside of avalanche-prone terrain to train the GPR model.

*Comment 6:*

The Discussion section could benefit from subheadings to improve readability.

**Response to comment 6:**

**We have added subheadings to the Discussion section.**

**Substantive line-by-line comments:**

Title: The title broadcasts the "leveraging of citizen science data" but no such data appear to be utilized in this study. I understand that the approach/findings of the paper have implications for guiding citizen science data collection, but I feel the title has potential to be misleading. Consider reframing.

**Response:**

**We have adjusted the title to better represent the subject of the work. It is now Leveraging Snow Probe Data, LiDAR and Machine Learning for Snow Depth Estimation in Complex Terrain Environments.**

Line 2. 'Address gaps in basin-scale snowpack modeling' is a bit vague. Can you briefly describe the gap you are addressing? Increasing spatial information of the snowpack?

**Response:**

**Line 2 of the abstract has been reworded to clarify the focus of the study.**

Line 9. Maybe state which dataset represents the "true snow depth distribution".

**Response:**

**We have changed this to reference the LiDAR snow depth distribution**

Line 12. Add "in the training data" after "excluded"

**Response:**

**We included this language.**

Line 34. Meromy et al., 2013 and Herbert et al., 2024 (references at end of document) are more recent papers which explore Snotel representativeness.

**Response:**

**Thank you for the suggested papers, we have reviewed and incorporated them.**

Line 37-39, 144. Is the assumption of normally distributed snow depth key to the methodology? As in, does the GPR make assumptions about the distribution of snow depth when making predictions? If yes, this assumption could be explored further in the discussion. Does the methodology deteriorate if snow depth is not normally distributed? If this is not key to the methodology, this information feels unnecessary. Additionally, it would be useful to show the lidar snow depth histogram to convince readers it follows a normal distribution.

**Response:**

**The GPR does rely on the target variable (snow depth) to be of a Gaussian distribution, and can lose robustness particularly for large outliers or heavy skewness. We justify the use of GPR based on the near normality of the LiDAR snow depth. We have added a figure (Figure 3, below) of the LiDAR snow depth distribution to section 2.2 and provided statistical justification for the assumption. Text has been added to the Discussion section to expand on the Gaussian assumption, and to explain this methodology may suffer when snow depth does not follow this assumption.**

**New Figure 3:**

[Figure]

**Figure 3.** Map and corresponding density distribution of the LiDAR-measured snow depth in Franklin Basin. The violin distribution of the snow depth is overlayed above a Gaussian distribution of 20000 randomly generated samples to express the near-normality of the snow depth distribution. Quartiles of the snow depth are shown as bold hashed lines, and the light grey hashed lines represent the Gaussian distribution quartiles.

Line 69: Well, not "anytime" (weather/clouds can still be a factor in the snow-free season. Consider rephrasing.
**Response:**
**We removed "anytime" to be more accurate.**

Line 70. I was a bit confused here when you mention 'snow-free lidar data'. If my understanding is correct, saying something like: 'physiographic data from snow-free lidar scans' would make this sentence easier to follow.
**Response:**
**We have reworded this sentence to improve clarity.**

Line 74 (paragraph). Citing examples of papers which use citizen science could be beneficial here. See Crumley et al. 2021.
**Response:**
**Thank you for the recommendation, we have included the reference.**

Figure 7. I recommend adding panels which show the modeled snow depth and lidar snow depth (in addition to the delta snow depth). This allows for the reader to make an easier visual comparison of the two maps. I also recommend adding sampling locations to the map.
**Response:**
**The recommended changes have been made to Figure 7 (see above).**

Figure 7. Is there an explanation for the horizontal blocks of similar error? There appear to be blocks of ~5 horizontal pixels that tend to register the same error (and maybe the same snow depth?). Is this an artifact of the GPR? Topography?
**Response:**
**This was an artifact of the plotting scheme. We have revised the pixel resolution of the plot to**

**better show the distribution.**

Line 132: How was it decided that northerly is used as the upwind barrier direction? Is this the dominant wind direction in the area?
**Response:**
**This assumption was based on prevailing wind direction observed at the nearby Logan Airport, Utah, provided by the Western Regional Climate Center. A citation has been included for this.**

Figures 2, 10, etc.: The use of red (FB) and green (HK) lines/markers may render some figure unreadable those with red-green color vision issues. Please consider revising to make the figures more accessible.
**Response:**
**We have adjusted the red and green color schemes to a more high contrast scheme of magenta and teal in Figures 2, 5, and 10.**

Line 142: Please provide the date of the sampling effort. I don't see it anywhere in the document.
**Response:**
**Text has been added to Sections 2.2 and 2.4 of the specific date of lidar and snow survey.**

Lines 142-148: Was ground-truthing conducted for the lidar-derived snow depths? If so, what differences/errors were found?
**Response:**
**The field snow sampling was only conducted in Hell's Kitchen, which was not covered by the LiDAR flight path. Text has been added to the Discussion section to addresses the potential for error and uncertainty from the lidar data.**

Line 147-148. The explanation for why one study area wasn't upscaled could be clarified.
**Response:**
**The Hell's kitchen Canyon DEM was not upscaled as this was the physical sampling subbasin, and the higher resolution was required to guide samplers to the relevant physiographic locations. The LiDAR snow depth and the physiographic rasters of the other regions were upscaled, as processing the data at 1.5 m resolution for the large region for multiple scenarios became computationally inefficient. Text has been added to section 2.3 to better explain this.**

Figure 3 and  Lines 134-138: One could argue that the avalanche runout zones (flatter slopes below the avalanche slopes) should be included as "avalanche-prone terrain", since the goal is to avoid measurements in dangerous zones.
**Response:**
**Thank you for this comment. This is true, however was beyond the scope for the current project. There is little available historical data of avalanche paths in the study region, and accurately defining the extent both horizontal and vertical, and the potential runout of an avalanche is very complex. We chose to maintain a simplified definition of avalanche-prone terrain with the aim to show the feature space may be masked without information loss in the model. This has been highlighted in Section 4.3 of the discussion**

Line 175-177. I don't see any results for the model which used five sampling locations.
**Response:**
**The full sensitivity analysis for the model from 5 sites to 200, as avy vs no avy has been included**

**in the supplemental materials (see Figure S1).**

Line 201. Any justification for the use of the GPR model? Pros/cons versus other models? Or just following the Oroza methodology?
**Response:**
**We investigated other algorithms such as Random Forest and XGboost, however the nature of these are too complex for the very small training data size of this study. The probabilistic GPR model is better suited to the small training set. This was the main reasoning for the GPR. A brief statement of this has been included in Section 2.5. A detailed comparison of algorithms was not in the scope of this work.**

Figures 7 & 8: It would be useful and interesting to show the maps of lidar snow depth and modeled snow depth in addition to the map of estimate errors (what is currently shown).
**Response:**
**We have adjusted Figure 8 (above) to display side-by-side the estimated snow depth for each subbasin result, as well as the respective errors. The full basin lidar snow depth has been presented in the new Figure 3 (above). Figure 7 (above) has also been updated with the recommendations.**

Figure 9. The left two plots are difficult to interpret based on the current colors. On the left I could barely find the curve with the lowest peak and in the middle plot I can only see four curves. Consider changing colors, using different line styles, or removing the fill on the curves.
**Response:**
**We have adjusted the color and line style scheme of figure 9 to improve clarity.**

Line 309: "streamflow forecasting" – it is odd that this is only mentioned in the conclusions but not earlier in the paper. Consider removing or providing more context earlier.
**Response:**
**We have removed this from the Conclusion.**

Line 310 (paragraph). In a similar vein to the avalanche terrain exclusions, I wonder how many appropriate sampling locations could be found. You currently choose the 'best' sampling location, but what if you selected the top 5 for each cluster? Then the samplers could select the location which is easiest to access. Maybe out of scope here, but just a thought!
**Response:**
**This is an interesting question for sure, and may be a focus of future work with this project. It was not included in the scope here, but thank you for the suggestion.**

Line 401. 'significant' has statistical implications. Maybe something like 'minimal losses' instead.
**Response:**
**We have incorporated this change.**

**Formatting and wording comments**

Lines 12, 29, and elsewhere: What does 'seamless' mean in these sentences?
**Response:**
**"Seamless" has been changed to "gridded" or "continuous" throughout to avoid confusion.**

Line 29: 'Products to produce' is a bit awkward.
**Response:**
**Replaced with "observations or techniques"**

Line 35: should be "snowpack" (no hyphen).
**Incorporated**

Line 37: Should be "snow depth" (no capital S)
**Incorporated**

Line 62: 'Aerial flown' is awkward. Maybe just 'aerial'.
**Incorporated**

Line 76. 'collected by such users via a mobile app platform'. Maybe 'reported' would be more appropriate.
**Incorporated**

Line 105: Careful with capitalization of cardinal directions here and in the rest of the document. No need to capitalize East, Easterly, etc.
**Incorporated**

Line 115: 'Snow-free' shouldn't be capitalized.
**Incorporated**

---

## Referee Report (RR1)

Liljestrand et al. have made significant improvements to the manuscript and satisfactorily addressed my comments. Below, I list some minor changes that should be addressed prior to publication. I do not need to see the manuscript again prior to acceptance by the journal.

65: High cost is also a significant drawback to lidar data. I would mention it in this paragraph.

82: Suggest deleting 'with the increase of users… remote snowpack information.' This seems unnecessary. It would be fine as "Thus, it is imperative to…"

85: This paragraph could be improved. The topic sentence should describe how SWE is the variable that is most important in snow sampling. Then you can describe how improved SD improves SWE. I would also more explicitly mention that SD is the variable you can reliably measure 10 times in a field day as well as the variable which is measured by lidar. As such you are mostly constrained to lidar. But, you could potentially model/extrapolate density to get estimates of SWE.

Line 90: 'broader' is vague, maybe 'more spatially extensive.'

153: Add the year to the date.

Fig. 3: Make the normal distribution lines thicker or change the color. Difficult to see.

187: Delete 'e.g. 12 hours.' Seems unnecessary.

339: Wording could be improved here. Do you mean: there is variation in performance when few samples are used, but the variation is limited with larger sample sizes?

Fig. 10: suggest add a line where y=0 to make the graph easier to interpret.

341: delete 'highly'

353. Suggest deleting the two commas surrounding 'or human error in the depth probe measurements.' Current iteration is clunky to read.

---

## Author Response (AR2)

**Manuscript Number:** Preprint egusphere-2024-3545
**Title:** Leveraging Snow Probe Data, LiDAR, and Machine Learning for Snow Depth Estimation in Complex Terrain Environments
**Subject:** Response to Manuscript Review Comments

Thank you again for your further comments and feedback on our manuscript. We have addressed the minor comments to improve figures, include suggested considerations and to improve clarity and readability. We hope the revisions adequately address the comments and remain open to further suggestions or questions.

**Response to Editor:**

*Comment 1:*

*My own suggestion is to improve the site locations in the topography shown in Fig. 5. Some of the sites appear to be obscured by the topographic lines. Could you please enhance that figure?*

**Response to comment 1:**

**Thank you for the comment, Figure 5 has been altered to better show the sites above the topographic lines.**

**Response to Reviewer #2:**

*Liljestrand et al. have made significant improvements to the manuscript and satisfactorily addressed my comments. Below, I list some minor changes that should be addressed prior to publication. I do not need to see the manuscript again prior to acceptance by the journal.*

*Comment 1:*

Line 65: High cost is also a significant drawback to lidar data. I would mention it in this paragraph.

**Response to comment 1:**

**The text has been adjusted to highlight the cost constraints of LiDAR**

*Comment 2:*

Line 82: Suggest deleting 'with the increase of users… remote snowpack information.' This seems unnecessary. It would be fine as "Thus, it is imperative to…"

**Response to comment 2:**

**Incorporated**

*Comment 3:*

Line 85: This paragraph could be improved. The topic sentence should describe how SWE is the variable that is most important in snow sampling. Then you can describe how improved SD improves SWE. I would also more explicitly mention that SD is the variable you can reliably measure 10 times in a field day as well as the variable which is measured by lidar. As such you are mostly constrained to lidar. But, you could potentially model/extrapolate density to get estimates of SWE.

**Response to comment 3:**

**Thank you for the suggestion. We have reworked this paragraph to better highlight SWE, as well as to better justify the application and benefit of snow depth in snow sampling.**

*Comment 4:*

Line 90: 'broader' is vague, maybe 'more spatially extensive.'

**Response to comment 4:**

**Incorporated**

*Comment 5:*

Line 153: Add the year to the date.

**Response to comment 5:**

**Incorporated**

*Comment 6:*

Fig. 3: Make the normal distribution lines thicker or change the color. Difficult to see.

**Response to comment 6:**

**Incorporated**

*Comment 7:*

Line 187: Delete 'e.g. 12 hours.' Seems unnecessary.

**Response to comment 7:**

**Incorporated**

*Comment 8:*

339: Wording could be improved here. Do you mean: there is variation in performance when few samples are used, but the variation is limited with larger sample sizes?

**Response to comment 8:**

**We have modified the sentence for clarity.**

*Comment 9:*

Fig. 10: suggest add a line where y=0 to make the graph easier to interpret.

**Response to comment 9:**

**Incorporated**

*Comment 10:*

341: delete 'highly'

**Response to comment 10:**

**Incorporated**

*Comment 11:*

Line 353. Suggest deleting the two commas surrounding 'or human error in the depth probe measurements.' Current iteration is clunky to read.

**Response to comment 11:**

**Incorporated**